DOI: 10.1038/s41467-018-07593-0 | OPEN

# Selective production of phase-separable product from a mixture of biomass-derived aqueous oxygenates

Yehong Wang [1], Mi Peng[2], Jian Zhang[1], Zhixin Zhang[1], Jinghua An[1,3], Shuyan Du[1], Hongyu An[1,3], Fengtao Fan[1], Xi Liu [4,5], Peng Zhai[2], Ding Ma [2] & Feng Wang[1]

Selective conversion of an aqueous solution of mixed oxygenates produced by biomass fermentation to a value-added single product is pivotal for commercially viable biomass utilization. However, the efficiency and selectivity of the transformation remains a great challenge. Herein, we present a strategy capable of transforming ~70% of carbon in an aqueous fermentation mixture (ABE: acetone–butanol–ethanol–water) to 4-heptanone (4-HPO), catalyzed by tin-doped ceria (Sn-ceria), with a selectivity as high as 86%. Water (up to 27 wt%), detrimental to the reported catalysts for ABE conversion, was beneficial for producing 4-HPO, highlighting the feasibility of the current reaction system. In a 300 h continuous reaction over 2 wt% Sn-ceria catalyst, the average 4-HPO selectivity is maintained at 85% with 50% conversion and >90% carbon balance. This strategy offers a route for highly efficient organic-carbon utilization, which can potentially integrate biological and chemical catalysis platforms for the robust and highly selective production of value-added chemicals.

[1] State Key Laboratory of Catalysis, Dalian National Laboratory for Clean Energy, Dalian Institute of Chemical Physics, Chinese Academy of Sciences, Dalian 116023, China. [2] College of Chemistry and Molecular Engineering and College of Engineering, BIC-ESAT, Peking University, Beijing 100871, China. [3] University of Chinese Academy of Sciences, Beijing 100049, China. [4] State Key Laboratory of Coal Conversion, Institute of Coal Chemistry, Chinese Academy of Sciences, Taiyuan, Shanxi 030001, China. [5] Syncat@Beijing, SynfuelsChina Co. Ltd, Beijing 101407, China. Correspondence and requests for materials should be addressed to D.M. (email: dma@pku.edu.cn) or F.W. (email: wangfeng@dicp.ac.cn)

With the development of sustainable energy and chemicals, interest in fermentation processes is reviving, because they can transform non-edible biomass feedstock to low-molecular-weight oxygenates, such as acetone, n-butanol, and ethanol in the acetone–butanol–ethanol–water (ABE) fermentation broth[1,2]. The crude fermentation broth contains low concentrations of oxygenates, which can be enriched by specific techniques such as pervaporation, gas stripping, and adsorption. However, the complexity/cost of the purification process limits its widespread practical applications. Thus, finding a method to directly and selectively convert crude aqueous oxygenate mixture to value-added chemicals, especially water-immiscible ones, thus allowing easy separation after reaction, is of great importance.

The efficiency and selectivity of the transformation process for biomass-derived intermediates remains a major techno-economic challenge[3]. Significantly, carbon–carbon coupling enables the conversion of small intermediate molecules to larger ones, which is critical for fuel production[4–7]. However, carbon–carbon cross-coupling between oxygenates is complicated. For homogeneous catalysts, although high yield can be achieved, the separation of the catalyst after reaction is difficult[8,9]. For heterogeneous catalysts, the selectivities for certain reactions are rather low[10] and both types of catalysts demand high purity of feedstock[11], which is a major obstacle to industrialization of biomass conversion. In ABE fermentation broth conversion, deprotonation of the C–H group at the α-positions of the primary products, including acetone, acetaldehyde, and butanal is not selective. Enolate intermediates generated in the process react with aldehyde or ketone to produce a randomly distributed ketone mixture with carbon number ranging from 5 to 11, which is ideal for fuel production but not enough for single chemical fabrication. Moreover, the presence of large amount of water deactivates the reported catalytic system in aqueous ABE solution[12]. It is pivotal to develop a highly efficient water-resistant catalyst for the direct conversion of crude aqueous oxygenate mixture.

Herein we report a simple reaction for the conversion of aqueous ABE fermentation broth, containing acetone, n-butanol, ethanol, acetic acid, and butyric acid to a water-immiscible product efficiently and selectively. Catalyzed by Sn-ceria catalyst, the biomass-derived oxygenates mixture was transformed to 4-heptanone (4-HPO) (a flexible precursor for the synthesis of various value-added chemicals or fuels (Supplementary Figure 1 and Supplementary Table 1)) with selectivity as high as 86%. A series of characterizations verified the importance of highly dispersed tin oxide and oxygen vacancies in the selective production

of 4-HPO from the ABE mixture. A biomass conversion route to transform low-quality biomass fermentation broth to easily separable value-added product was constructed using this strategy, thereby greatly reducing process complexity.

## Results and Discussion

**Catalytic results.** First, a mixture of acetone, n-butanol, ethanol, and water in a f 9:51:1:22 ratio (by weight) was fed into a vertical fixed-bed reactor loaded with a ceria catalyst, which was prepared by a precipitation method[13]. Liquid products were collected using a cold-trap. Carbon balance, conversion, yield, and selectivity were calculated based on carbon number of each molecule, as per the following equations (see Supplementary Figure 2 and Supplementary Table 2).

$$\text{Carbon balance (\%)} = \frac{n(c)_{gas} + n(c)_{liquid}}{n(c)_{feed}} \times 100\% \quad (1)$$

$$\text{Conversion (\%)} =$$
$$\frac{n(c)_{feed} - n(c)_{liquid,\,butanol} - n(c)_{liquid,\,ethanol} - n(c)_{liquid,\,acetone}}{n(c)_{feed}} \times 100\% \quad (2)$$

$$\text{Yield (\%)} = \frac{n(c)_{4-HPO}}{n(c)_{feed}} \times 100\% \quad (3)$$

$$\text{Selectivity } (4-HPO)(\%) =$$
$$\frac{n(c)_{4-HPO}}{n(c)_{gas} + n(c)_{liquid} - n(c)_{liquid,\,butanol} - n(c)_{liquid,\,ethanol} - n(c)_{liquid,\,acetone}} \times 100\% \quad (4)$$

where $n(c)_{4-HPO}$ is the number of moles of carbon atoms in 4-HPO, $n(c)_{feed}$ is the number of moles of carbon atoms in the feedstock, and $n(c)_{liquid}$ is the number of moles of carbon atoms in the liquid trapped in the tank, including products and unreacted feedstock. The terms $n(c)_{liquid,\,butanol}$, $n(c)_{liquid,\,acetone}$, and $n(c)_{liquid,\,ethanol}$ represent the number of moles of carbon atoms of n-butanol, acetone, and ethanol, respectively, trapped in the liquid. The terms $n(c)_{gas}$ is the number of moles of carbon atoms in gaseous products.

When the reaction was conducted at 400–420 °C over ceria catalyst, the liquid-phase product naturally separates into two layers: the light-yellow upper layer is the oil phase and the colorless bottom layer is the aqueous phase. The oil phase was found to consist of 78% 4-HPO and 15% 2-pentanone, whereas in the aqueous phase mainly unreacted acetone and water were detected. In the gas phase, < 4% $CO_2$ was detected, demonstrating that most of the converted reactants have

**Table 1 Conversion of aqueous ABE solution over various catalysts[a]**

| Catalyst | C balance (%) | Conv. (%) | 4-HPO yield (%) | Liquid product distribution (%) | | | | |
|---|---|---|---|---|---|---|---|---|
| | | | | 4-HPO | PNO | BAL | 2-HPO | Others |
| _ | 100 | – | – | – | – | – | – | – |
| Ceria | 95 | 22 | 14 | 78 | 15 | 1 | 2 | 4 |
| Sn-Ceria | 106 | 71 | 61 | 86 | 3 | 3 | 3 | 5 |
| Zn-Ceria | 100 | 95 | 59 | 73 | 27 | – | – | – |
| Fe-Ceria | 92 | 95 | 57 | 75 | 25 | – | – | – |
| In-Ceria | 97 | 50 | 44 | 94 | 6 | – | – | – |
| Sn-Ceria[b] | 85 | 81 | 55 | 68 | 23 | – | 9 | – |
| Sn-Ceria[c] | 108 | 82 | 59 | 72 | 22 | – | 5 | 1 |
| Sn-Ceria[d] | 105 | 72 | 45 | 63 | 29 | – | 6 | 2 |

[a] Reaction conditions: 3.2 g of catalyst (40–60 mesh), pretreated in H2 (15 mL min⁻¹) at 420 °C for 1 h and then the reaction was conducted at 420 °C for 2 h; the A:B:E weight ratio is 9:51:1, the water content is 27 wt%, weight hourly space velocities (WHSV) = 0.5 h⁻¹; N2 as carrier gas (flow rate = 10 mL min⁻¹), the content of Zn, Fe, or In doped ceria is 2 wt%. [b] A:B:E weight ratio is 3:6:1, the water content is 21 wt%. [c] The A:B:E weight ratio is 1:5.7:1.2, the water content is 20 wt%. [d] The A:B:E weight ratio is 1.7:4:1, the water content is 20 wt%. BAL butanal, BBA butyl butyrate, 2-HPO 2-heptanone, 4-HPO 4-heptanone, PNO 2-pentanone

been fixed into liquid products. For comparison, no product was obtained in the uncatalyzed reaction (Table 1).

**Reaction pathways**. The reaction with pure oxygenate mixture without water was studied. Interestingly, the selectivity of 4-HPO decreased from 84% to 31% together with a decrease of conversion from 22% to 10%, suggesting that water is critical for the

highly selective synthesis of 4-HPO in this system. (Fig. 1(a, b)). This also holds when using pure $n$-butanol or $n$-butanol/water as the feed (Supplementary Figure 3). Significantly, with $n$-butanol/water as the feed, 4-HPO was proposed to be produced by the following steps (Supplementary Figure 4): (i) dehydrogenation of $n$-butanol to butanal and hydrogen, (ii) esterification of butanol and butanal to butyl butyrate and hydrogen, (iii) reduction of

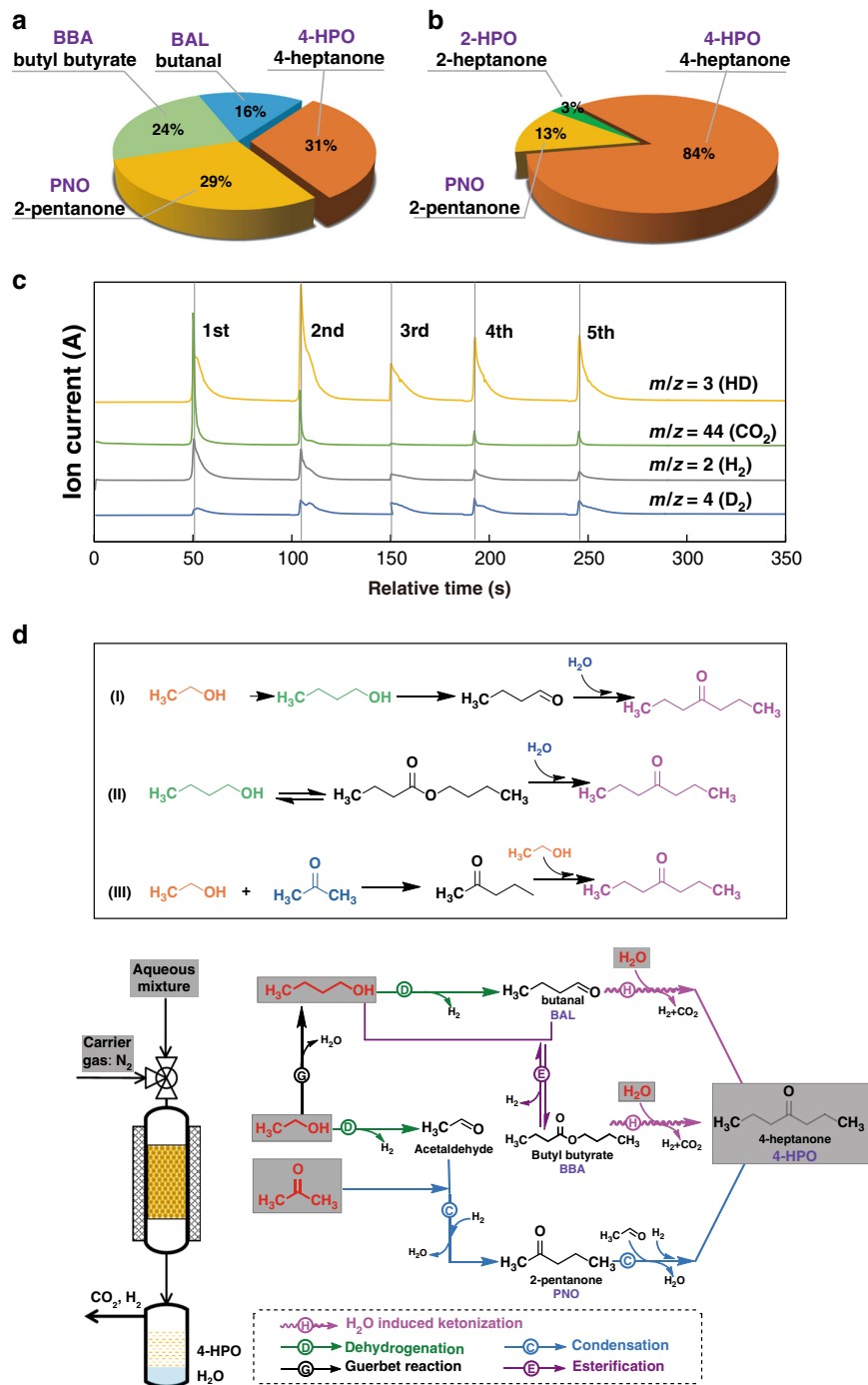

**Fig. 1** The catalytic function of water in ABE aqueous solution conversion reaction and proposed reaction pathways. **a** ABE as feedstock (A:B:E weight ratio is 9:51:1). **b** ABE with water as feedstock (A:B:E:H$_2$O weight ratio is 9:51:1:22). Reaction conditions: ceria (16.0 g, 14–25 mesh), N$_2$ as carrier gas (flow rate = 33 mL min$^{-1}$), 400 °C, WHSV = 0.5 h$^{-1}$. **c** Pulse reaction of a mixture of $n$-butanol and D$_2$O (detected by on-line mass spectrometry). Ceria (1.0 g, 14–25 mesh), 10 μL per injection of $n$-butanol and D$_2$O mixture, Ar as carrier gas (flow rate = 30 mL min$^{-1}$), 400 °C. The fluctuation of each pulse is due to non-uniform sampling of butanol for each injection (from butanol/D$_2$O emulsion mixture). **d** Proposed reaction pathways leading to 4-HPO in a fixed-bed reactor with acetone, $n$-butanol, ethanol, and water as feedstock over ceria-based catalyst

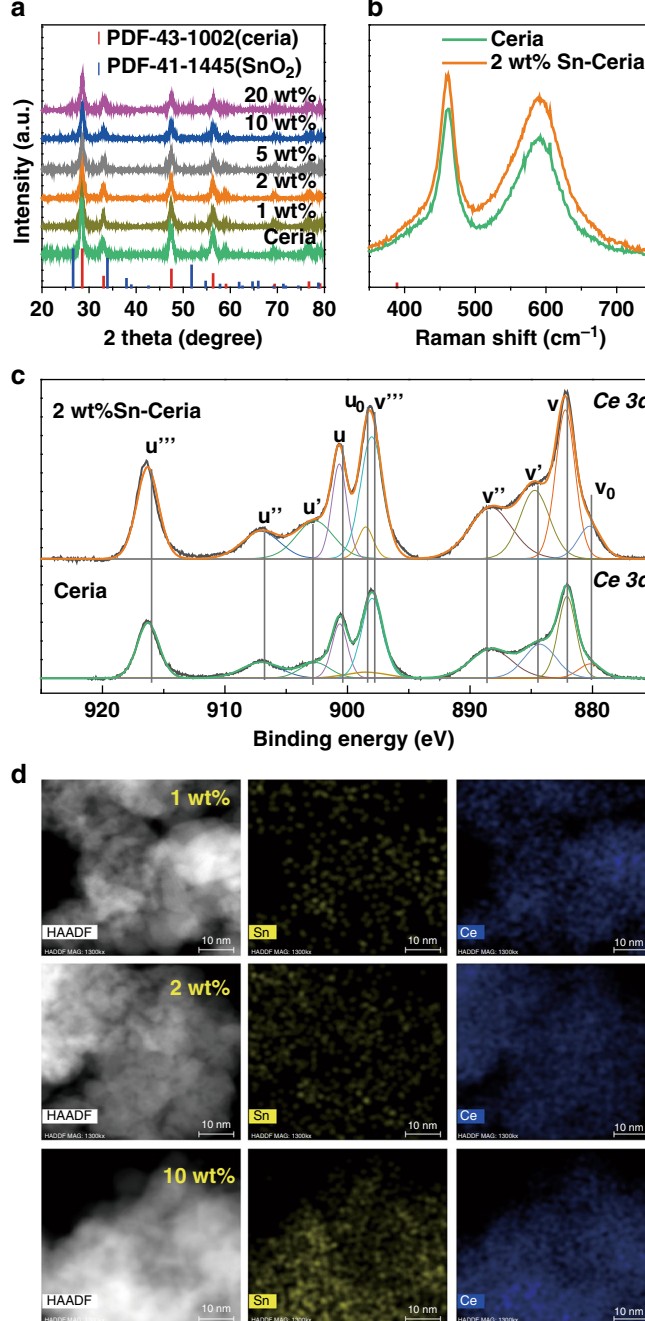

**Fig. 2** Characterizations of ceria and Sn-ceria catalysts. **a** XRD patterns (using Cu Kα radiation) of ceria and Sn-ceria catalysts with various Sn loading ranging from 1 wt% to 20 wt%. **b** UV-Raman spectra of ceria (green line) and 2 wt% Sn-Ceria (orange line). **c** XPS spectra for Ce 3d core level of pristine ceria and 2 wt% Sn-ceria. **d** TEM and EDS elemental mapping images (Ce and Sn) of 1 wt%, 2 wt%, and 5 wt% Sn-ceria catalysts. Scale bar, 10 nm

ceria by hydrogen to produce oxygen vacancies, (iv) water dissociation on the oxygen vacancies to produce active oxygen ($O^*$) and hydrogen, (v) aldol condensation of butanal to hydroxyaldehyde, (vi) reaction of hydroxyaldehyde with $O^*$ to form surface-adsorbed carboxylate, and finally, (vii) dehydrogenation and decarboxylation of the carboxylate to obtain 4-HPO[14]. In order to confirm the role of water in the oxygenate conversion process, isotope labeling experiments were conducted over ceria at 400 °C. Upon injecting $D_2O$ to the pretreated ceria in $H_2$ at

400 °C, hydrogen deuteride and deuterium (HD and $D_2$) are observed (Supplementary Figure 5), indicating that the water dissociates at the oxygen vacancies on the ceria surface to produce hydrogen and heal the defects[15,16]. When a mixture of *n*-butanol and $D_2O$ (5 mL and 1 mL, respectively) was injected and carried to the catalyst bed by Ar gas, the formation of $H_2$ (m/z = 2), HD (m/z = 3), $D_2$ (m/z = 4), and $CO_2$ (m/z = 44) was immediately detected by on-line mass spectrometry, which confirms the role of water in the transformation of *n*-butanol. In all of the conducted five pulses, HD and $D_2$ were detected (Fig. 1(c)).

To elucidate the reaction pathways for forming 4-HPO, ceria-catalyzed reactions with different mixed oxygenate substrates were conducted. For the substrate without *n*-butanol, condensation of acetone and ethanol can produce 4-HPO over ceria. The feed containing only ethanol generated 38% acetone, 31% 2-pentanone, 20% 2-pentanol, and 11% *n*-butanol in the liquid[17]. The condensation of 2-pentanone and ethanol selectively generated 4-HPO, confirmed by co-feeding 2-pentanone, and ethanol (Supplementary Figure 6). In addition, the feed containing only acetone resulted in self-condensation products of acetone, including 15% mesityl oxide, 7% isophorone, and the further isomerized (6% 4-methylpent-4-en-2-one) or hydrogenated products (5% 4-methylpentan-2-one). 3, 5-Dimethylphenol (26%), formed via demethanation of isophorone, and its derivative (13% 2, 4, 6-trimethylphenol), were also observed in the liquid phase[18–20]. However, this complex product distribution was not observed for ABE mixture conversion, suggesting that under our reaction condition, the condensation of acetone with ethanol is more favorable than the self-condensation of acetone. Acetone is a product of acetic acid ketonization reaction over acid-base catalysts[21].

Interestingly, although ceria is a versatile catalyst for dehydrogenation, alcohol Guerbet reaction, condensation, and esterification reactions, all these reactions generate 4-HPO with high selectivity. Figure 1(d) illustrates the complicated reaction network with the feedstock of acetone, *n*-butanol, ethanol, and water, which could be summarized as follows:

(1) *n*-Butanol can be dehydrogenated to butanal, which reacts with water on reduced ceria catalyst to form 4-HPO, as confirmed by the result of co-feeding of butanal and water (Supplementary Figure 7).

(2) Besides acting as reactant, *n*-butanol is also a product generated via ethanol Guerbet reaction, from which 4-HPO was produced via route (1).

(3) *n*-Butanol can be dehydrogenated to butanal, and then to butyl butyrate via esterification of *n*-butanol and butanal. The reaction of butyl butyrate and water generates 4-HPO, experimentally confirmed by the co-feeding of butyl butyrate and water (Supplementary Figure 8).

(4) Dehydrogenation of ethanol produces acetaldehyde, which then condenses with acetone to give 2-pentanone, whose further condensation with ethanol produces 4-HPO.

**Catalyst synthesis and characterization.** From the above discussion, it is noteworthy that aldehydes produced by the dehydrogenation of ethanol and *n*-butanol are important precursors for selective 4-HPO generation. Promoters were introduced to ceria, to further enhance the dehydrogenation ability, including a series of metal oxides such as MgO, ZnO, $Fe_2O_3$, and $SnO_2$. In the *n*-butanol dehydrogenation reaction (Supplementary Figure 9 (a)), butanal, which is a key intermediate for 4-HPO generation, was the major product obtained over these metal oxide catalysts, among which $SnO_2$ offers the best selectivity toward butanal (over 99%). We then prepared a series of Sn-modified ceria catalysts with different approaches (marked A, B, C, and D) and employed

them in aqueous ABE conversion (Supplementary Figure 9 (b)). Among these catalysts, the Sn-doped ceria catalyst (marked as Sn-ceria) prepared by co-precipitation (style D), showed the highest selectivity toward 4-HPO and the best catalytic stability (Supplementary Figure 9 (c)), the optimal loading for Sn being 2 wt% (Supplementary Figure 10). For comparison, 2 wt% In-, Fe-, and Zn-doped ceria catalyts were also prepared with the similar method. All of them showed less efficient catalytic performances with either moderate conversion or lower selectivity of 4-HPO (Table 1). Ceria can provide redox sites especially oxygen vacancies, for the condensation, esterification, and Guerbet reactions, whereas tin is proposed to promote catalytic hydrogen removal. We conducted X-ray diffraction (XRD), inductively coupled plasma (ICP) analysis, $N_2$ adsorption–desorption, and transmission electron microscopy (TEM) experiments to characterize the best Sn-ceria catalysts (Fig. 2, Supplementary Figure 11-13, Supplementary Table 3-5). From the XRD profile, only the cubic phase of ceria is observed with Sn loading ≤ 5 wt% (please note the 5 wt% means that the weight loading of Sn added to the ceria catalyst is 5 wt%, whereas the loading amounts of the corresponding catalysts, determined by ICP, can be found in Supplementary Table 3). In contrast, the $SnO_2$ phase appeared when 10 wt% and 20 wt% Sn was doped into the ceria catalysts (Fig. 2(a)). This indicates that when the loading of Sn is not too high, the tin species are highly dispersed over the ceria surface. The highly dispersed state of the tin species on 2 wt% Sn-ceria catalyst is further confirmed by TEM observations and energy dispersive spectroscopic elemental mappings, which show that the tin species are almost homogeneously distributed over the ceria surface and no aggregation or sintering of Sn is identified (Fig. 2(d)). This highly dispersed tin species may accelerate the dehydrogenation of the complex reaction network of ABE mixture transformation and drive the reaction towards 4-HPO. Furthermore, UV-Raman spectra are found to be dominated by the strong $F_{2g}$ mode of the ceria fluorite phase at 462 cm$^{-1}$ and a band at 592 cm$^{-1}$ due to the defect-induced (D) mode for both ceria and 2 wt% Sn-ceria (Fig. 2(b)). The relative intensity ratio of $I_D/I_{F_{2g}}$ (concentration of oxygen vacancy) is about 0.86 for 2 wt% Sn-ceria, which is a little higher than that for ceria (0.81), indicating that Sn addition promoted the formation of oxygen vacancies, as confirmed by the higher concentration of Ce(III) in 2 wt% Sn-ceria than pristine ceria (32.8% vs. 23.3%, Fig. 2(c)). Both the highly dispersed tin species and oxygen vacancies are critical for the good catalytic performance obtained.

**Effects of feedstock**. Different bacteria and fermentation technologies generate ABE solutions with different concentrations

and compositions[1]. Thus, it is imperative to know the effect of water content on this reaction process. We conducted the reaction using three artificial aqueous ABE solutions with different water contents (623 g L$^{-1}$, 707 g L$^{-1}$, and 783 g L$^{-1}$), selected based on the representative water content that can be obtained by current separation techniques reported in literature (Supplementary Figure 14). In all the cases, the selectivity towards 4-HPO is higher than 80%. Apart from substrate concentration, the ratio of different oxygenates might also have an impact on the reaction. We tested an aqueous ABE solution with different reactant ratios (e.g.: A:B:E mass ratio = 3:6:1, water content: ~20 wt%, to simulate fermentation by *Clostridium acetobutylicum*) and observed that 4-HPO selectivity of ~68% and yield of ~55% can be obtained, suggesting the pervasiveness of the current method (Table 1). In addition, organic acids, mainly acetic acid and butyric acid, are usually produced during ABE fermentation[2,22]. Here, the addition of acetic acid and butyric acid (1.7 wt% and 1.5 wt%, respectively) increased 4-HPO selectivity from 84% to 90% (Supplementary Figure 15).

**Catalytic stability**. To test the catalyst stability, we performed a continuous reaction by using 2 wt% Sn-ceria (Fig. 3(a)). Initially, 4-HPO was produced with ~70% conversion and ~86% selectivity. After about 300 h, the average 4-HPO selectivity was maintained at 85% with 50% conversion and > 90% carbon balance. The cold-trapped liquid phase naturally separates into two layers (Fig. 3(b)), easily separable in a separatory funnel. Phase (A) contains 4-HPO, 2-pentanone, unreacted acetone, and *n*-butanol, while phase (B) mainly contains unreacted acetone and water (Supplementary Figure Figure 16). High purity of 4-HPO (>95%) can be obtained by fractional distillation of phase (A), showing the power of current process (Supplementary Figure 17). After 300 h of reaction, extensive shrinkage was observed in the catalyst surface area, possibly due to carbonaceous deposition, which are the main reasons for the slight deactivation of the catalyst (Supplementary Figure 18-20). Developing highly stable tin–ceria-based catalyst system is the next goal of this biomass transformation process.

In summary, herein we have developed a strategy for the catalytic conversion of aqueous ABE solutions to phase-separable 4-HPO with ~86% selectivity and ~70% conversion over Sn-ceria catalyst. The presence of water was critical for this highly selective catalytic system. Sn-modified ceria serves as a versatile catalyst for the complicated catalytic reaction network and all the catalyzed reactions yield 4-HPO with high selectivity. This system paves a path for effective biomass conversion to obtain value-added products.

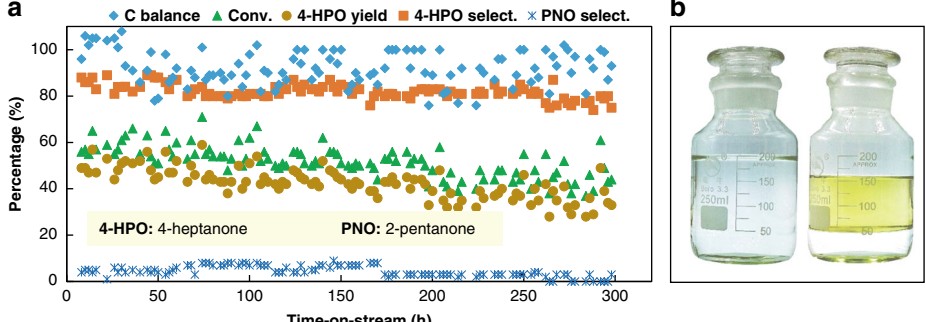

**Fig. 3** Stability test of 2 wt% Sn-ceria catalyst in aqueous ABE solution conversion reaction. **a** Catalytic stability test result. Catalyst (3.2 g, 40–60 mesh), $N_2$ as carrier gas (10 mL min$^{-1}$), 420 °C, WHSV = 0.5 h$^{-1}$. ABE solution as feedstock with A:B:E:water ratio of 9:51:1:22 (by weight). **b** Images of ABE feedstock (left) and trapped liquid products (right)

## Data availability

All data are available from the corresponding author upon reasonable request.

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

## Acknowledgements

This work received financial support from the "Strategic Priority Research Program of the Chinese Academy of Sciences" (XDB17000000, XDB17020300), the National Key R&D Program of China (2017YFB0602200), and the Natural Science Foundation of China (21721004, 21690080, 21711530020, 21725301, 91645115, 21473003, 21673273, 21872163, and 21821004).

## Author contributions

F.W. and D.M. designed the study. Y.W., J.Z. and Z.Z. performed most of the reactions. Y.W., M.P. and J.A. did the data analysis. Y.W., M.P., D.M. and F.W. wrote the paper. S.D., H.A., F.F., and X.L. conducted the experiments of XPS, Raman, and TEM. P.Z. provided reagents and revised the paper.

## Additional information

**Competing interests:** The authors declare no competing interests.

