## [Peer Review File · Nature Communications]

Reviewers' Comments:

Reviewer #1:

Remarks to the Author:

The authors have done a very good job addressing all of the reviewers comments and the manuscript has improved significantly but I still question whether the relevance and impact warrants publication in Nature Communications. Maybe a journal such as ChemSusChem, Sustainable Energy and Fuels might be more appropriate based on the current outlook for ABE fermentation.

Reviewer #2:

Remarks to the Author:

The authors report a process for conversion of mixed oxygenate stream typically obtained after fermentation (containing acetone, butanol, ethanol and water (ABE)) to 4-heptanone (4-HPO). The authors show that Sn-Ceria catalyst is active as well as selective for the conversion of ABE to 4-HPO. The authors show that Sn-Ceria is also stable under the reaction condition. The study is done methodically, however, there are some inconsistency in the reported data and as such the manuscript cannot be accepted for publication. The authors should address the following points.

1. The authors show that the catalyst is stable over 300 hours on stream in Fig. 3, however, it is observed from figure S10 that under similar reaction condition (i.e., using same reaction temperature and same feedstock composition) the catalyst deactivates rapidly. It is observed that the yield decreases from 50% to 20% in 5 hours and since the selectivity remains constant implying that conversion decreases with time on stream. The data reported are not consistent. Authors should carefully look at this inconsistency.
2. For the second step of the two step process the feed consists of 2-pentanone and ethanol in the molar ratio of 1:2. Thus there are two moles of ethanol per mole of 2-pentanone. From Fig S6 (c), it is observed that all ethanol is consumed however, there are no products to account for that loss. Authors should comment on the product distribution and carbon balance during this reaction.
3. In Fig 1(c), the CO₂ peak decreases with injection. Authors do not discuss this observation. An explanation for this observation should be provided.
4. The authors discuss the two-stage reaction with acetic acid and ethanol (molar ratio 1:1) as feedstock. This discussion is confusing. It appears from the discussion in main text and Fig. S6(a) that two-steps were conducted in one setup, where the product of reactor 1 is the feed of reactor 2 with an additional ethanol stream, however, the caption indicate that two separate reactions were conducted, one with ethanol and acetic acid (1:1) as feed and another as PNO and ethanol (1:2) as feed. If the two reactions were not conducted in series than the discussion doesn't add any new insight to the paper.
5. On Pg-4, Line 111-113. The authors the equilibrium is shifted the product side as the product are separated in oil phase, however The reaction is conducted at 420oC, at this temperature there is only gas phase, thus this explanation for improved conversion is not true
6. The authors should report the conversion level for Fig 1 (a,b) and for figure S3.
7. There is no discussion about Fe-Ceria, Zn-Ceria, and In-Ceria in the manuscript. Particularly, In-Ceria has better selectivity to 4-HPO than Sn-Ceria. Why was Sn-Ceria chosen?
8. Fig S4 should also include BBA formation pathway.
9. Last three rows in Table 1, shows the conversion and yield at different A:B:E ratios. However, the authors do not report the water concentration. Since it is shown subsequently that water is important for selective production of 4-HPO, the authors should include the concentration of water in the feed for all experiments.
10. The authors claim that condensation of ethanol and 2-pentanone generates 4-HPO which is confirmed by co-feeding the reactants, but, there is no data reported to support this claim.

11. Authors should report metal loading in Table 1.
12. Authors should add conversion figure in S10.
13. "full-organic carbon utilization" is misleading as CO₂ and other unidentified products are formed. Authors should rephrase the last sentence in the abstract.
14. The manuscript must be carefully edited for typographical and grammatical errors. For example, a complete paragraph is repeated in the manuscript (Page 5, left column, line 36-56).

Reviewer #3:

Remarks to the Author:

All the comments I raised about the previous submission have been responded by the authors in an appropriate manner. Moreover, the revised statements have been supported by additional experimental data along with catalyst characterization data. The abstract and concluding remark are also consistent to the results and discussion in the revised manuscript.

Response to the Reviewers' Comments:

Reviewer #1:

Comment: The authors have done a very good job addressing all of the reviewers comments and the manuscript has improved significantly but I still question whether the relevance and impact warrants publication in Nature Communications. Maybe a journal such as ChemSusChem, Sustainable Energy and Fuels might be more appropriate based on the current outlook for ABE fermentation.

Response: We appreciate the reviewer's comment. However, we believe Nature Communication is a better platform for the readers to know this nice work, which has provided a simple and straightforward method to convert biomass into high value products through a combination of bio-chem method.

Reviewer #2:

Comment 1: The authors show that the catalyst is stable over 300 hours on stream in Fig. 3, however, it is observed from figure S10 that under similar reaction condition (i.e., using same reaction temperature and same feedstock composition) the catalyst deactivates rapidly. It is observed that the yield decreases from 50% to 20% in 5 hours and since the selectivity remains constant implying that conversion decreases with time on stream. The data reported are not consistent. Authors should carefully look at this inconsistency.

Response 1: We thank the reviewer for the nice suggestion. The yield of 4-HPO is ranging from 40% to 60% during the life test (Fig. 3), which is higher than that shown in Fig. S10. This is majorly due to different reaction conditions. The catalyst life test in Figure 3 was conducted with a lower WHSV (0.5 h^{-1}), larger amount of catalyst (3.2 g), and a lower flow rate of carrier gas (nitrogen gas $10 \text{ mL}\cdot\text{min}^{-1}$) compared with that in Fig. S10 (0.8 h^{-1} , 2 g, $33 \text{ mL}\cdot\text{min}^{-1}$), although the other conditions such as reaction temperature and feedstocks remain the same. The slower flow rate of carrier gas leads to a larger contact time with the catalyst (6 second for Fig. 3 and 1.5 second for Fig. S10), which improves conversion. Also, we need to point out due to the liquid injection nature of feedstock, there is some variation in the product yield in short period of time which is also observed in the Figure 3A. However, the long-term

stability is relatively good. Following the reviewer's comment, we added the conversion of these Sn-ceria catalysts (1%, 2%, 5%, 10%, 20% Sn loading) into the revised manuscript.

Comment 2: For the second step of the two step process the feed consists of 2-pentanone and ethanol in the molar ratio of 1:2. Thus there are two moles of ethanol per mole of 2-pentanone. From Fig S6 (c), it is observed that all ethanol is consumed however, there are no products to account for that loss. Authors should comment on the product distribution and carbon balance during this reaction.

Response 2: As shown in Fig.S6(c), ethanol is converted to 4-HPO. First, acetaldehyde is generated via ethanol dehydrogenation. Then, the condensation of acetaldehyde and 2-pentanone forms 4-HPO. In addition, acetone is generated as by-product via the ketonization of acetaldehyde. The selectivity of 4-HPO is ranging from 77% ~ 84% with carbon balance >85%. Since this part a little bit diverges the reader's attention, after considering the reviewer's opinions (and comment 4), we have removed this part from the revised manuscript.

Comment 3: In Fig 1(c), the CO₂ peak decreases with injection. Authors do not discuss this observation. An explanation for this observation should be provided.

Response 3: We thank the reviewer for the suggestion. The formation of CO₂ is possibly due to the intermediates transformation or even steam reforming of butanol. However, the feedstock here for injection is an emulsion mixture of butanol and water due to the insolubility of butanol in water. This leads to the non-uniform sampling of butanol for each injection, and therefore uneven butanol/D₂O ratios in each injection. We believe that the instability of the pulse reaction can be resulted. Surely we just want to show qualitative instead of quantitative result in the manuscript, which is used to illustrate the ability of the catalyst in activating water as well as *n*-butanol.

Following the reviewer's suggestion, we have added a sentence in the revised caption of Figure 1C "The fluctuation of each pulse is due to non-uniform sampling of butanol for each injection (from butanol/D₂O emulsion mixture)".

Comment 4: The authors discuss the two-stage reaction with acetic acid and ethanol (molar ratio 1:1) as feedstock. This discussion is confusing. It appears from the discussion in main text and Fig. S6(a) that two-steps were conducted in one setup, where the product of reactor 1 is the feed of reactor 2 with an additional ethanol

stream, however, the caption indicate that two separate reactions were conducted, one with ethanol and acetic acid (1:1) as feed and another as PNO and ethanol (1:2) as feed. If the two reactions were not conducted in series than the discussion doesn't add any new insight to the paper.

Response 4: We agree with your suggestion and delete this part.

Comment 5: On Pg-4, Line 111-113. The authors the equilibrium is shifted the product side as the product are separated in oil phase, however The reaction is conducted at 420°C, at this temperature there is only gas phase, thus this explanation for improved conversion is not true.

Response 5: Thanks very much for pointing out this mistake. We have removed this over-statement in the revised manuscript.

Comment 6: The authors should report the conversion level for Fig 1 (a,b) and for figure S3.

Response 6: The conversions for Fig.1(a) and (b) is 10 % and 22%, respectively, and 14% and 27% for Fig.S3 (a) and (b), respectively. It has been added into the revised manuscript.

Comment 7: There is no discussion about Fe-Ceria, Zn-Ceria, and In-Ceria in the manuscript. Particularly, In-Ceria has better selectivity to 4-HPO than Sn-Ceria. Why was Sn-Ceria chosen?

Response 7: We have added the discussion about the catalytic performance of Fe-Ceria, Zn-Ceria, and In-Ceria in the revised manuscript. For In-Ceria catalyst, the conversion is moderate (50%), although the selectivity of 4-HPO is high (94%). This is the best result after experimental optimizations over In-Ceria. The lower catalytic activity of In-Ceria is due to its lower concentration of oxygen vacancy (In-Ceria: $I_D/I_{F2g} = 0.72$ vs Sn-Ceria: $I_D/I_{F2g} = 0.86$ from UV-Raman). Thus, due to its excellent conversion and selectivity, Sn-Ceria was chosen in this manuscript. We have added the explanation in the revised manuscript.

Comment 8: Fig S4 should also include BBA formation pathway.

Response 8: The BBA formation pathway have been added into the revised manuscript and highlighted in yellow. It is shown as follows:

Comment 9: Last three rows in Table 1, shows the conversion and yield at different A:B:E ratios. However, the authors do not report the water concentration. Since it is shown subsequently that water is important for selective production of 4-HPO, the authors should include the concentration of water in the feed for all experiments.

Response 9: The water content is 21 wt%, 20 wt% and 20 wt% for [b], [c] and [d], respectively. We have added these data into Table 1 in the revised manuscript and they are highlighted in yellow.

Comment 10: The authors claim that condensation of ethanol and 2-pentanone generates 4-HPO which is confirmed by co-feeding the reactants, but, there is no data reported to support this claim.

Response 10: Sorry for missing this point. We had conducted the reaction with co-feeding ethanol and 2-pentanone. The results show that this condensation could generate 4-HPO with ca. 80% selectivity. These data are supplied in the revised Supplementary Materials as Fig. S6 and highlighted in yellow.

Fig. R2. (a) Yield and selectivity of 4-HPO during the reaction of co-feeding 2-pentanone and ethanol; (b) A typical GC result of this reaction. Reaction conditions: ceria (3.2 g, 40-60 mesh), the mole ratio of 2-pentanone to ethanol is 1:2, N₂ as carrier gas (10 mL·min⁻¹), 420 °C, WHSV= 0.5 h⁻¹.

Comment 11: Authors should report metal loading in Table 1.

Response 11: These data had been supplied in the part of “catalyst preparation”. The content of metal oxides in doped ceria was 2% (wt/wt) relative to the ceria support. For clarification, we add these data into Table 1 in the revised manuscript and highlighted in yellow.

Comment 12: Authors should add conversion figure in S10.

Response 12: We add the conversion into Fig. S10 in the revised manuscript.

Comment 13: “full-organic carbon utilization” is misleading as CO₂ and other unidentified products are formed. Authors should rephrase the last sentence in the abstract.

Response 13: We thank the reviewer for the reminding. The last sentence in the

abstract has been corrected to “This strategy offers a route for highly efficient organic-carbon utilization, which can potentially integrate biological and chemical catalysis platforms for the robust and highly selective production of value-added chemicals.”

Comment 14: The manuscript must be carefully edited for typographical and grammatical errors. For example, a complete paragraph is repeated in the manuscript (Page 5, left column, line 36-56).

Response 14: Following your advice, we check the manuscript carefully. No repeated paragraph is found in the manuscript. In addition, we have made some corrections in the revised manuscript and they are highlighted in yellow.

Reviewer #3:

All the comments I raised about the previous submission have been responded by the authors in an appropriate manner. Moreover, the revised statements have been supported by additional experimental data along with catalyst characterization data. The abstract and concluding remark are also consistent to the results and discussion in the revised manuscript.

Response: We thank the reviewer for the nice recommendation.

Reviewers' Comments:

Reviewer #2:

Remarks to the Author:

The authors have addressed our previous comments.